# The AMP deaminase of the mollusk *Helix pomatia* is an unexpected member of the adenosine deaminase-related growth factor (ADGF) family

**George Tzertzinis, Mehul B. Ganatra, Cristian Ruse¤, Christopher H. Taron,
Bryce Causey, Liang Wang, Ira Schildkraut** *

New England Biolabs, Ipswich, MA, United States of America

¤ Current address: Moderna Therapeutics, Cambridge, MA, United States of America
* schildkraut@neb.com

**Data Availability Statement:** The raw and processed sequencing data generated in this study have been submitted to the NCBI BioProject

## Abstract

We report here the first occurrence of an adenosine deaminase-related growth factor (ADGF) that deaminates adenosine 5' monophosphate (AMP) in preference to adenosine. The ADGFs are a group of secreted deaminases found throughout the animal kingdom that affect the extracellular concentration of adenosine by converting it to inosine. The AMP deaminase studied here was first isolated and biochemically characterized from the roman snail *Helix pomatia* in 1983. Determination of the amino acid sequence of the AMP deaminase enabled sequence comparisons to protein databases and revealed it as a member of the ADGF family. Cloning and expression of its cDNA in *Pichia pastoris* allowed the comparison of the biochemical characteristics of the native and recombinant forms of the enzyme and confirmed they correspond to the previously reported activity. Uncharacteristically, the *H. pomatia* AMP deaminase was determined to be dissimilar to the AMP deaminase family by sequence comparison while demonstrating similarity to the ADGFs despite having AMP as its preferred substrate rather than adenosine.

## Introduction

Adenosine 5' monophosphate (AMP) deaminases catalyze the conversion of AMP to inosine 5' monophosphate (IMP) and ammonia (Fig 1). The AMP deaminases (AMPD) are classified as a subgroup of the adenyl-deaminase family [1] based on their amino acid sequence similarity. The other members of the adenyl-deaminase family belong to the adenosine deaminase (ADA), adenine deaminase (ADE), adenosine deaminase-like (ADAL), and adenosine deaminase-related growth factors (ADGF) subgroups.

The foot muscle of the mollusk, *H. pomatia*, is one of the richest sources (units per mg protein) of AMP deaminase (HPAMPD) [2]. HPAMPD was found to convert 5' AMP, 5' ADP, 5' ATP, and NADH to their respective inosine derivatives with low micromolar $K_m$ values [3]. These characteristics are unlike those of the animal skeletal muscle AMP deaminases, which

database (https://www.ncbi.nlm.nih.gov/bioproject/) under accession number PRJNA936131.

**Funding:** This work was funded internally by New England Biolabs, Ipswich, MA, of which all authors are employees. The funder provided support in the form of salaries for: G.T., M.B.G., C.R., C.H.T., B.C., L. W., and I.S., but did not have any additional role in the study design, data collection and analysis, decision to publish, or preparation of the manuscript. The specific roles of these authors are articulated in the 'author contributions' section.

**Competing interests:** I have read the journal's policy and the authors of this manuscript have the following competing interests. George Tzertzinis, Mehul B. Ganatra, Cristian Ruse, Christopher H. Taron, Bryce Causey, Liang Wang, Ira Schildkraut are/were employees of New England Biolabs, a manufacturer and vendor of molecular biology reagents. New England Biolabs funded the work and paid the salaries of all authors. There are no patents, products in development or marketed products to declare. This does not alter our adherence to PLOS ONE policies on sharing data and materials.

have a strict specificity for 5' AMP and relatively high $K_m$ values, >1 mM. Our initial interest in the HPAMPD came from its high activity and its ability to deaminate a di-adenosine tetra-phosphate (AppppA) to di-inosine tetraphosphate (IppppI) [4].

For this report we purified HPAMPD and subsequently cloned, sequenced and expressed the cDNA encoding the HPAMPD. We found its amino acid sequence to be most similar to an adenosine deaminase from another mollusk, *Aplysia californica*. The sea slug, *A. californica*, encodes a biologically and biochemically characterized adenosine deaminase which was initially designated as mollusk derived growth factor (MDGF) [5] and later, as a member of the ADGF subfamily [1]. The ADGF subfamily is characterized by secreted deaminases which convert the extracellular nucleoside adenosine, a signaling molecule, to inosine. This subfamily of adenyl deaminases is represented in humans as the enzyme encoded by the ADA2 (also known as CECR1) gene [6]. ADA2 is involved in a developmental disorder known as Cat Eye Syndrome [7]. We show, by sequence similarity, that the AMPD from *H. pomatia* is a member of the ADGF subfamily of adenyl deaminases despite the fact that its preferred substrate is AMP. This is the first report of a member of the ADGF subfamily of adenyl-deaminases to prefer the nucleotide AMP over the nucleoside adenosine.

## Results

In furthering the characterization of the AMP deaminase of *H. pomatia* we first isolated the native protein from *H. pomatia* foot muscle and determined the sequences of the peptides that comprise it by 2D LC-MS/MS [8, 9]. Using these peptide sequences, we identified the ORF with the highest peptide sequence coverage from a *H. pomatia* transcriptome. The cDNA which encoded the protein was identified, expressed and the resulting recombinant enzyme characterized.

### Purification and characterization of the AMP deaminase from *H. pomatia*

The native HPAMPD was purified from the foot muscle from *H. pomatia* as described in Materials and methods. Deamination was measured by change in absorbance at either $OD_{265}$ or $OD_{260}$ [10]. The activity of the HPAMPD for the deamination of the nucleotides AMP, ATP, and adenosine, to their inosine equivalents is shown in Fig 2. The rate of deamination of adenosine and ATP is 0.04 and 0.25, respectively, relative to the rate of AMP. The specific activity of the native HPAMPD preparation was 125 units per mg of protein. However, the protein preparation appeared by PAGE analysis (S1 Fig in S1 File) to comprise a protein of 60 kD and contaminating proteins of 32 and 27 kD. This result is consistent with the monomer

**Fig 1. AMP deaminase reaction scheme.**

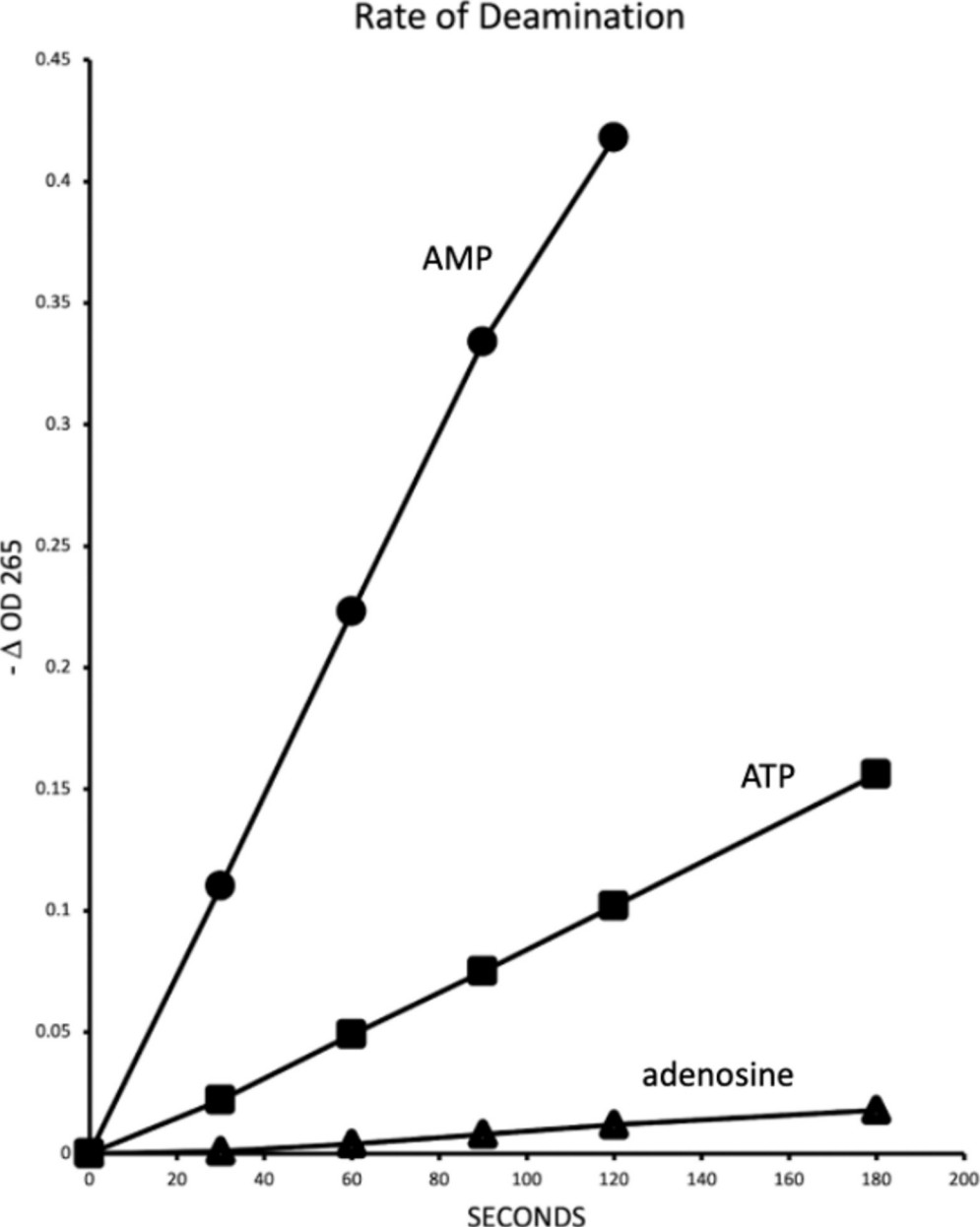

**Fig 2. Relative rates of deamination.** Activity of native *H. pomatia* AMP deaminase on 100 µM 5' AMP, 5' ATP, and adenosine. The reaction volume was 0.4 ml. The change in absorbance at $OD_{265}$ was followed in a spectrophotometer at 25˚ C.

size reported by Stankiewicz [2] at 60–64 kD with an additional two proteolytic degradation products of the monomer observed here. The specific activity reported by Stankiewicz [11] was similar at 117 units per mg protein.

## Peptide analysis by mass spectrometry

The purified HPAMPD protein was analyzed using a two-dimensional LC MS/MS system (see Materials and methods). The *de novo* peptides derived from the mass spectrometry were

```
RSVVRMSQVGAVTMVSIICVVVLGAVGGPVAGLAVRFPTMDEYTNAREELIGSEQYLRVG   60
GSITLNDKEKKLNQFILREKRAIIENSRLNKTQYIPAVSFFLSKSQMESTPIFKIIKDMP  120
KGAALHLHDTASARIDWIVSNATYRDHVYMCMDRDNFVRLTVSGTGLPADSGCEWKLVET  180
ERANSGDIAAFDHWLKSNISLLTTDPLVTYPSLDKVWGRFDKHFSQLRGIIYHTPIRRDY  240
YRQILEEFRSDNVQYVEVRSSLSGYYDLDGTVHDPEYGLQLYKAVTEEFVRTYPDFSGAK  300
IIKSTARVKPNTDVFNDVKLSMDLYKRYPGFFLGFDLVAQEDPNTPLLGYIDSLLYPSRQ  360
NPPVSLPYYFHAGETNWQGTEVDYNLVDALLLNATRIGHGFALIKHPRVIELVKSRGVAV  420
EVNPVSNQLLGLVKDLRNHAAAPLLAQNVPVVISSDAPGVWEALPMSHDMYVAFMDLVGE  480
DAGLDVLKQLVWNSIQYSSMNATEKRTALKLLQAKWNKFINDSLIKWKLTNKKVIG      536
```

**Fig 3. The amino acid sequence of the *H. pomatia* transcriptome contig 71391.** The sequence segments covered by LC-MS/MS derived peptides are shown in blue text.

intersected with open reading frames (ORFs) that were obtained by translating the *H. pomatia* transcriptome. Peptide scoring validation was based on q<0.01 [12]. Using the peptide sequences, we identified the ORF from the transcriptome with the highest peptide sequence coverage. The peptides covered 67% of a 536 amino acid ORF encoded by contig 71391 (Fig 3). The list of identified peptides from analyses of the digests is detailed in Supporting Information (S1 Table).

In order to determine the sequence and obtain a clone for expressing the protein in a recombinant form we generated cDNA from *H. pomatia* RNA. The cDNA was used as a template for PCR amplification using primers designed based on the sequence of contig 71391. The resulting 1596 bp fragment was sequenced and inserted into the pET28c vector (see Material and methods). Several nucleotide positions in the cDNA differed from the transcriptome which resulted in nine amino acid differences from contig 71391 presumably because the transcriptome had been derived from a different population of *H. pomatia* (S2 Fig in S1 File).

The contig 71391 encodes for a protein with an amino terminal end consistent with a signal peptide as predicted by SignalP-4.1 [13] of MSQVGAVTMVSIICVVVLGAVGGPVAG which *in vivo* should result in a secreted protein with either an amino terminal sequence of AVGGPV AGLAVRFPT or LAVRFPT. Both alternative amino terminal versions of the processed protein were submitted to structure prediction by ColabFold [14]. The resulting structures indicated that in the protein with the additional 8 terminal residues, AVGGPVAG, did not appear to be a structural feature of consequence (S3 Fig in S1 File).

## HPAMPD is a member of the ADGF subfamily of adenyl-deaminases

The generation of a phylogenetic tree based on amino acid sequence reveals HPAMPD to cluster within the ADGF subfamily of the adenyl-deaminase family (Fig 4). Consistent with the ADGF subfamily, the signal peptide of HPAMPD is a characteristic indicating the protein is secreted. InterPro [15] analysis of the HPAMPD amino acid sequence also indicated it is a member of the adenosine deaminase-related growth factors (ADGF). The determination that the HPAMPD was a member of the ADGF subfamily of adenosine deaminases rather than a member of the AMPD subfamily was surprising because of its preference for AMP over that of adenosine as substrate.

Observing that HPAMPD is a member of the ADGF subfamily, we used MAFFT [17] to align HPAMPD with representatives of the ADGF subfamily (Fig 5) which includes the most similar member, *A. californica* ADGF, with a protein sequence identity of 57% and an additional mollusk, *Lottia gigantea*. The other members span the vertebrates. Fortuitously, there is an experimentally derived three-dimensional structure for the human ADGF [18], with a protein sequence identity of 38%. This allowed us to superimpose it with the predicted HPAMPD structure.

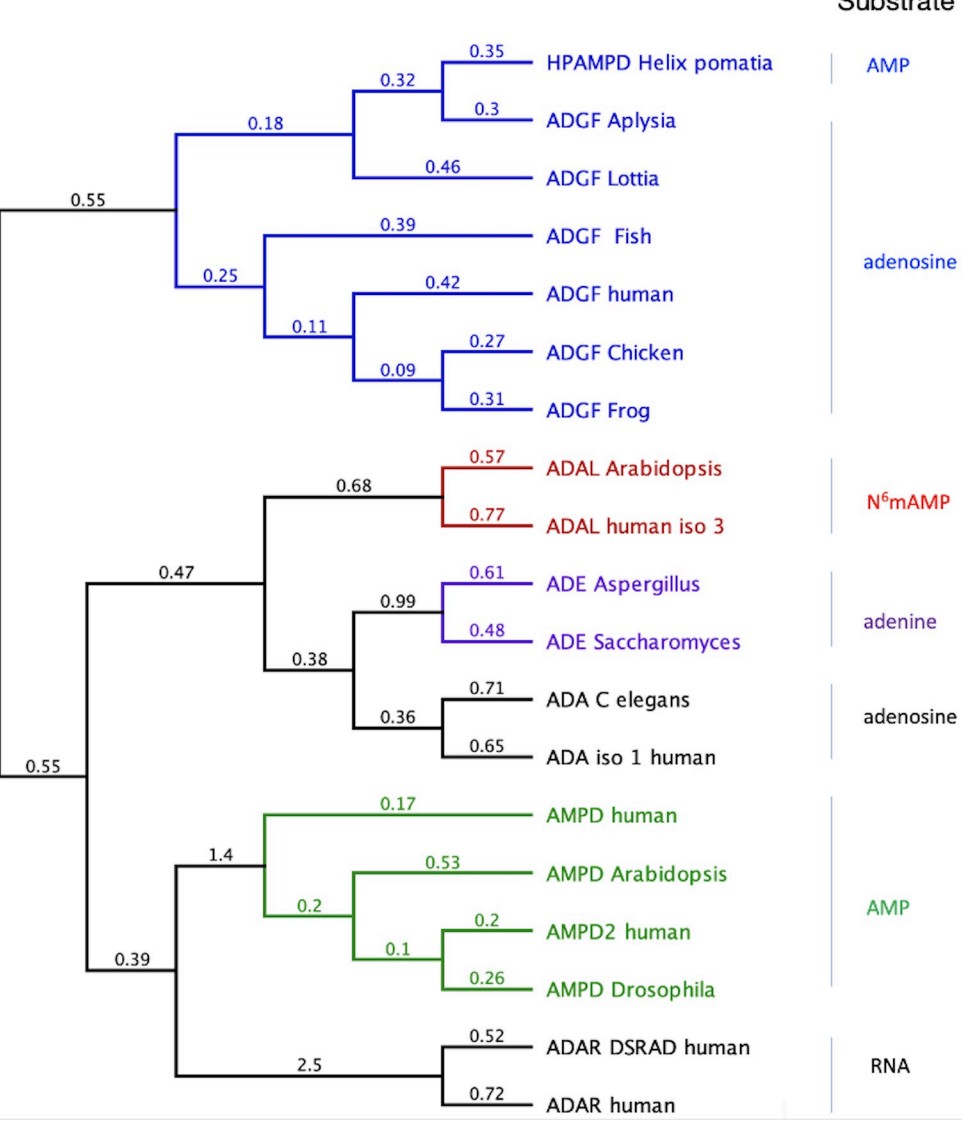

**Fig 4. Phylogenetic tree of the adenyl-deaminase family of proteins.** The tree was generated with FastTree [16] from a MAFFT alignment. The numerical values at each branch represent substitutions per site. ADGF are the adenosine deaminase-related growth factors, AMPD are the AMP deaminases, ADE are the adenine deaminases, ADAL are the adenosine deaminase-like proteins, ADAR are the adenosine deaminases acting on RNA, and the ADA are the adenosine deaminases. The substrate assigned to each subfamily is indicated on the right. The accession numbers for each enzyme follow: ADGF human, AAF65941.1; ADGF Aplysia, AAD13112.1; ADGF Fish, AF384217.1; ADGF Lottia, XP_009053965.1; ADGF Chicken, AAX10953.1; ADGF Frog, AAX10952.1; ADAL Arabidopsis, NP_192397.2; ADAL human iso3, NP_001311295.1; ADE Aspergillus, AL56636.1; ADE Saccharomyces, NP_014258.1; ADA C elegans, NP_872091.1; ADA human iso1, NP_000013.2; AMPD human, NP_000471.1; AMPD Arabidopsis, NP_565886.1; AMPD2 human, NP_001244290.1; AMPD Drosophila, NP_727740.2.

## Structure comparison of HPAMPD to human and *A. californica* ADGF

Structures of the HPAMPD and *A. californica* ADGF were predicted with AlphaFold2 [14, 20] and were compared to the reported experimentally derived human ADGF structure, 3LGG, [18] obtained from the Protein Data Bank (PDB) [21]. The PDB files were visualized and aligned with Pymol (The PyMOL Molecular Graphics System, Version 2.0 Schrödinger, LLC.)

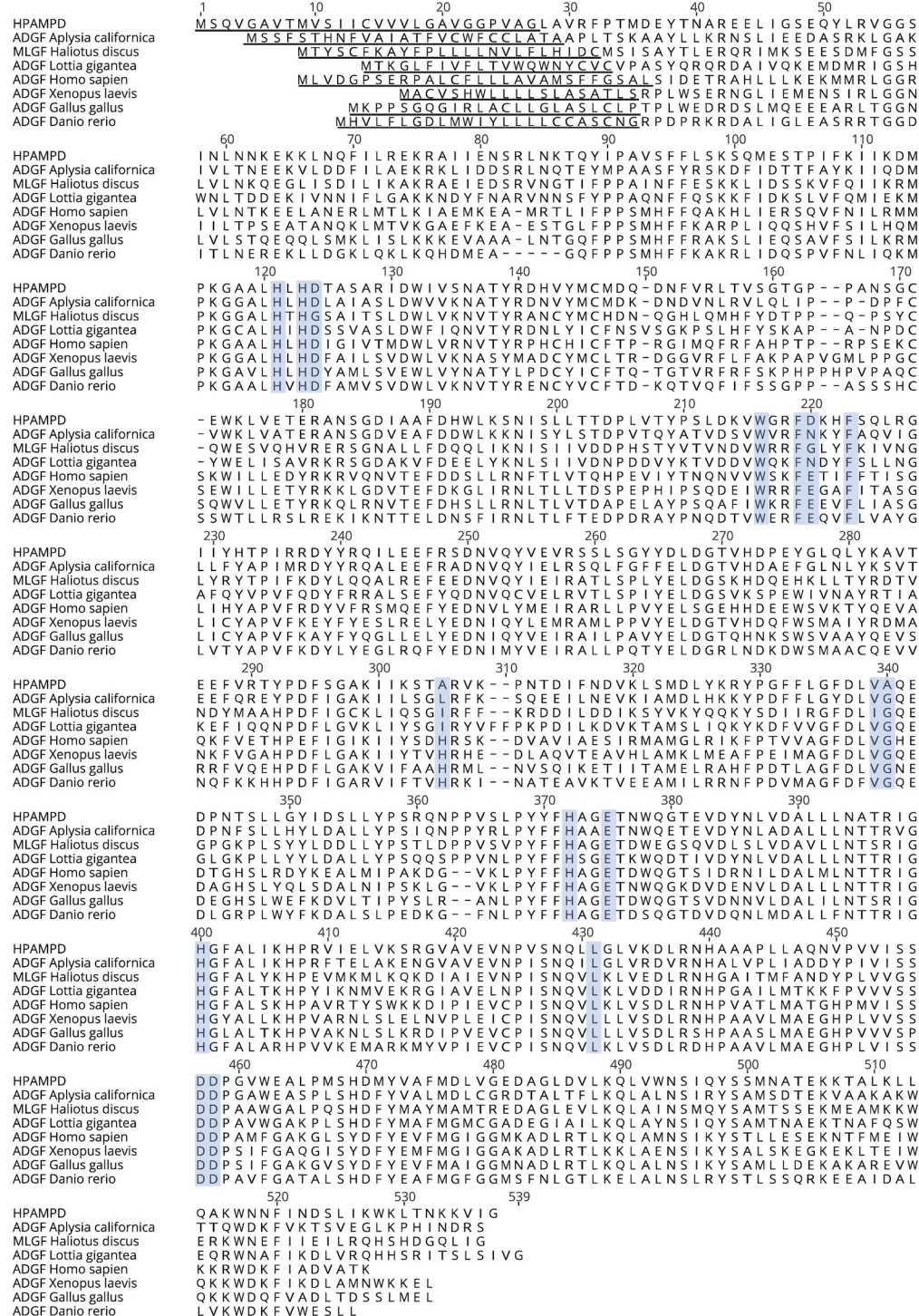

**Fig 5. Alignment of the HPAMPD to other ADGF subfamily members.** Underlined residues are predicted signal sequences [19] and blue outlined boxes are catalytic residue positions defined by [18] in human ADA2. HPAMPD sequence is derived from cDNA.

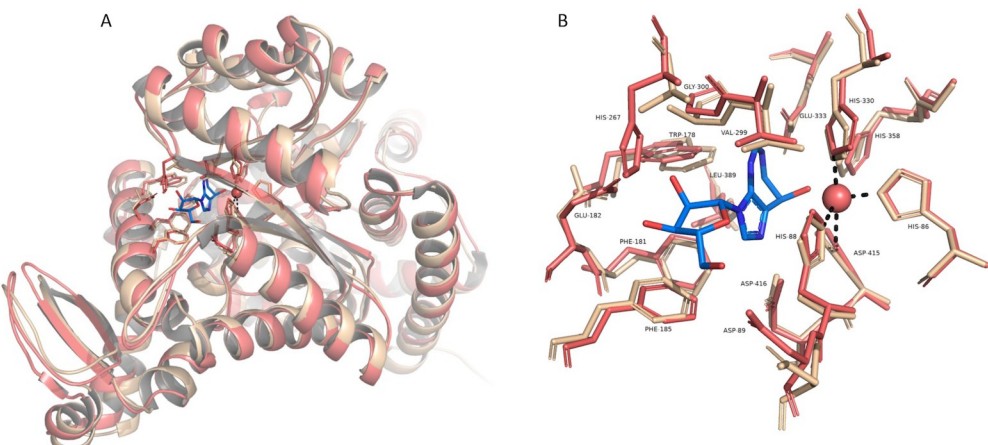

**Fig 6. Structural Comparison of HPAMPD and human ADGF.** (A) A superimposition of the AlphaFold2 predicted HPAMPD monomer and a monomer of the human ADGF (PDB 3LGG). Human ADGF is colored pink and the HPAMPD is colored cream. Side chains of active site residues and coformycin (colored blue) are drawn as sticks. The zinc ion is represented as a sphere and dashed lines depict interactions with coordinating residues from human ADGF. (B) A close-up view of the active site residues, zinc and coformycin. The labeled residues correspond to the 3LGG structure.

(Fig 6A). HPAMPD is aligned to *A. californica* ADGF with a RMSD of 0.594 and to human ADGF with an RMSD of 0.777. Structure alignments of the PDB files with Pymol demonstrate the close structural relatedness of HPAMPD to the two ADGFs. The active site of the human ADGF is elucidated because it formed a cocrystal with coformycin, a transition state analog and potent inhibitor of adenosine deaminases. Based on the structure of the human ADGF, which harbors a tightly coordinated zinc ion at the base of its active site [18] it is expected that the HPAMPD also contains a zinc ion as all four of the zinc-coordinating residues of the human sequence are conserved and aligned in the HPAMPD structure (Fig 6B). Furthermore, Alphafold2 predicted that HPAMPD is a homodimer, as is the experimentally derived structure for the human ADGF [18]. The extensive dimer interactions described for the human ADGF are also present in the predicted homodimer of HPAMPD (S3 Fig in S1 File).

## Cloning the cDNA encoding HPAMPD for expression in *Pichia pastoris*

Because the ADGFs as a group are extracellular eukaryotic secreted proteins, we utilized the budding yeast *Pichia pastoris* expression system [22] to express and secrete HPAMPD (see Materials and methods). The processed amino terminus of the HPAMPD is engineered to start at residues LAVR by use of the N-terminal secretion signal from *S. cerevisiae* alpha-factor. The carboxy end is terminated with 10 amino acid histidine residues. The amino acid sequence of the protein as secreted from *P. pastoris* is shown in Fig 7. The molecular weight of the protein as expressed in *P. pastoris* is 58676 daltons which is consistent with the value from SDS-PAGE of 60 kD (S1 Fig in S1 File). The specific activity of the recombinant HPAMPD preparation was determined to be 550 units/mg of protein.

## Characterization of recombinant HPAMPD activity

The recombinant form of the HPAMPD has a similar spectrum of activity on adenyl substrates as the native enzyme preparation from this study and the enzyme characterized in the 1984 Stankiewicz paper [2] (Table 1). All three 5' nucleotides, AMP, ADP and AMP are actively deaminated while the absence of a phosphate at the 5' position lowers the activity substantially.

```
LAVRFPTMDEYTNAREELIGSEQYLRVGGSINLNNKEKKLNQFILREKRAIIENSRLNKT  60
QYIPAVSFFLSKSQMESTPIFKIIKDMPKGAALHLHDTASARIDWIVSNATYRDHVYMCM  120
DQDNFVRLTVSGTGPPANSGCEWKLVETERANSGDIAAFDHWLKSNISLLTTDPLVTYPS  180
LDKVWGRFDKHFSQLRGIIYHTPIRRDYYRQILEEFRSDNVQYVEVRSSLSGYYDLDGTV  240
HDPEYGLQLYKAVTEEFVRTYPDFSGAKIIKSTARVKPNTDIFNDVKLSMDLYKRYPGFF  300
LGFDLVAQEDPNTSLLGYIDSLLYPSRQNPPVSLPYYFHAGETNWQGTEVDYNLVDALLL  360
NATRIGHGFALIKHPRVIELVKSRGVAVEVNPVSNQLLGLVKDLRNHAAAPLLAQNVPVV  420
ISSDDPGVWEALPMSHDMYVAFMDLVGEDAGLDVLKQLVWNSIQYSSMNATEKKTALKLL  480
QAKWNNFINDSLIKWKLTNKKVIGHHHHHHHHH                             514
```

**Fig 7. The amino acid sequence of the HPAMPD cDNA clone expressed in *P. pastoris*.**

Adenosine, 3'5'-cyclic AMP and 3' AMP are poor substrates. This profile is consistent with the presumption that the native and recombinant forms of HPAMPD are one and the same enzyme and represent the same enzyme characterized by Stankiewicz in 1983 and 1984 [2, 3].

Kinetic constants for recombinant HPAMPD and AMP, $K_m$ and $K_{cat}$, were determined from linear initial rates of reaction where 57.5 fmoles of enzyme were incubated in 200 μL reactions with various concentrations of AMP. The $K_m$ of 25 μM and $K_{cat}$ (turnover number) of 170 per second are derived from the Lineweaver-Burk plot (Fig 8). Stankiewicz [3] reported a $K_m$ value of 31 μM for 5' AMP.

Preincubation of the enzyme at 65° C in the absence of substrate in 10 mM Tris-HCl pH 7.5, 50 mM NaCl, 1 mM EDTA for 30 minutes reduces its activity by more than 95%, while 42° C preincubation has no effect on activity (S4 Fig in S1 File). Analysis of thermal denaturation with Prometheus (Nanotemper) indicates the HPAMPD initiates denaturation at 47° C and melts (50% unfolded) at 54° C.

## Comparison of *A. californica* and human ADGF activities on adenosine and AMP

The determination that the HPAMPD was a member of the ADGF subfamily of adenosine deaminases was surprising because of its preference for AMP over that of adenosine as substrate. This led us to question the annotation of ADGFs, as we could find only two biochemical evaluations of the lack of deamination of AMP [23, 24]. Therefore, we directly compared the activities of HPAMPD, A. *californica* and human homologues on both adenosine and AMP. The relative initial rates of the reactions are listed in Table 2. The results substantiate the assertion that the human [24] and A. *californica* ADGFs deaminate adenosine and do not deaminate AMP, whereas HPAMPD deaminates AMP and does not significantly deaminate adenosine.

**Table 1. Relative rates of deamination.** Spectrophotometric assay of deamination of adenyl base monitored by loss of $OD_{260}$ or $OD_{265}$. The rates are linear initial rates and expressed as a percentage of the 5' AMP rate. Reactions were at 25° C in 20 mM Tris-HCl pH 7.5, 50 mM NaCl and the indicated concentration of each substrate.

| Substrate | Relative activity from (Stankiewicz 1984) 70 μM substrate | Relative activity of native 100 μM substrate | Relative activity of recombinant 100 μM substrate |
|---|---|---|---|
| 5' AMP | 100 | 100 | 100 |
| 5' ADP | 95 | 83 | 74 |
| 5' ATP | 36 | 25 | ND |
| 3'5'-cyclic AMP | 0.35 | 2 | 0.5 |
| adenosine | 10 (@3 mM adenosine) | 4 | 2.8 |
| 3'-5' adenosine diphosphate (PAP) | 8 (@ 3mM PAP) | 1.6 | 0.7 |
| 3' AMP | ND | 0.9 | < 0.1 |

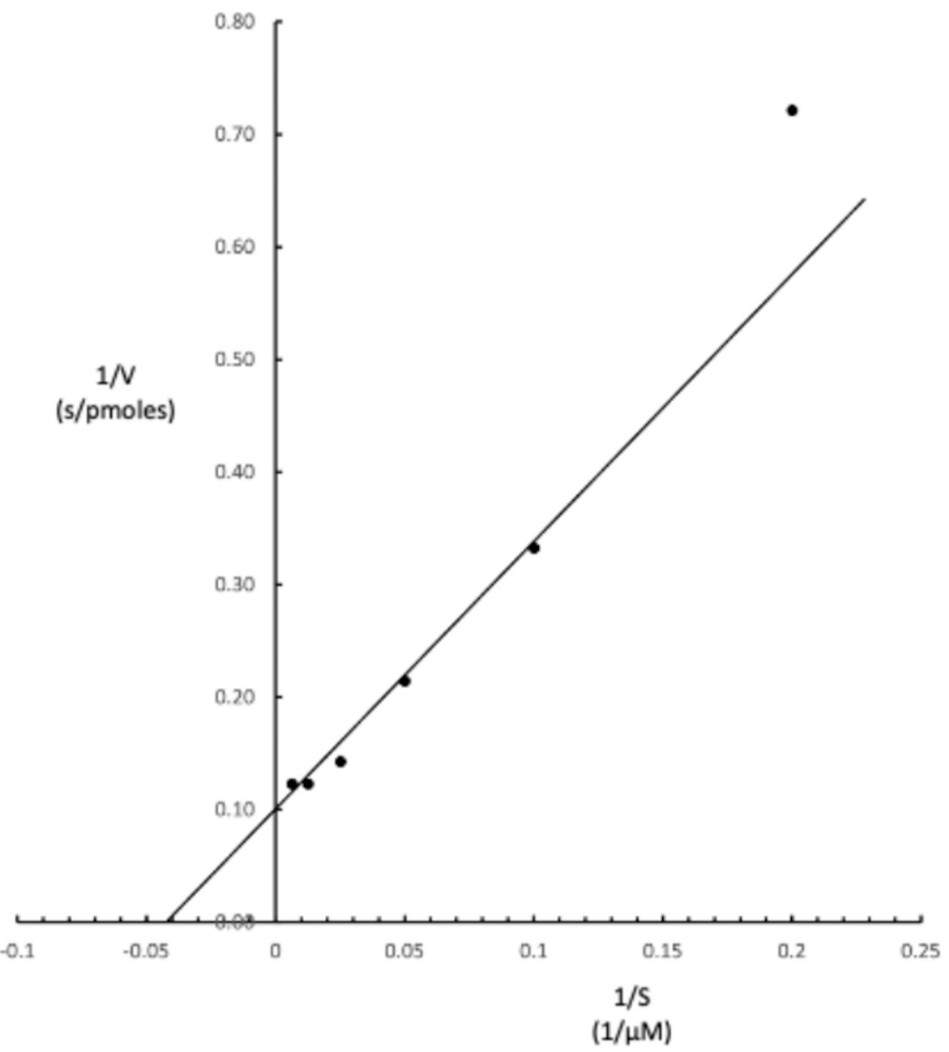

**Fig 8. Lineweaver-Burke plot for determination of $V_{max}$ and $K_m$ values for recombinant HPAMPD with AMP.**

## Discussion

Here, we report for the first time the identification of an ADGF family member whose substrate preference is adenosine 5' monophosphate. HPAMPD deaminates AMP with a $K_m$ of 25 μM and a $K_{cat}$ of 170 per second. Within the ADGF family, the $K_m$ of human ADGF for

**Table 2. Relative initial rates of deamination of adenosine and AMP for each deaminase.** The deamination was measured by loss of $OD_{265}$ in 10 mM Tris-HCl pH 7.5, 1 mM EDTA, 50 mM NaCl buffer. The substrate concentrations were 100 μM for the HPAMPD and *A. californica* enzymes and 2 mM for the human enzyme. The reaction temperature for *H. pomatia* and *A. californica* ADGFs was 25˚ and 37˚ for human ADGF. All values were obtained with recombinant enzymes except for the value in parentheses which was obtained with the native preparation of HPAMPD.

|  | AMP | Adenosine |
|---|---|---|
| HPAMPD | 1 (1) | 0.03 (0.04) |
| *Aplysia* ADGF | 0.001 | 1 |
| Human ADGF | < 0.01 | 1 |

adenosine is 2 mM [6], while those for most other ADGF's are lower, the lowest reported is from an invertebrate ADGF with a $K_m$ value for adenosine of 18 μM [25]. On the other hand the AMP deaminases (EC 3.5.4.6) listed at www.brenda-enzymes.org [26] have $K_m$ values for AMP ranging from 223 μM to 13 mM. The ten-fold lower $K_m$ of HPAMPD is a kinetic property that distinguishes it from the archetype AMP deaminase family.

We present evidence that the AMP deaminase cloned here from *H. pomatia* is the same as the one reported in 1983 [3]. HPAMPD is unlikely to be the only ADGF which prefers AMP. The few ADGFs which have been biochemically examined have been shown to deaminate adenosine. Here we demonstrate that the ADGFs most closely related to HPAMPD, the *A. californica* ADGF and the human ADGF prefer adenosine to AMP as substrate. ADGFs listed in protein databases are annotated as such primarily based on sequence similarity. Prediction of gene annotations ideally should utilize full descriptions of the encoded proteins' biochemical properties, that is in the case of an enzyme, its substrates and those related compounds that are not [27]. However, in practice, that is understandably an extremely difficult requirement to fulfill. Among the 1799 ADGF family members listed in the UniProtKB database (Uniprot consortium) only 7 members have been manually annotated by Swiss-Prot. Of the manually annotated members only 4 have been assayed for activity on adenosine and only one of those on AMP for which there was less than 1% of the activity as there was on adenosine [23]. Here we identified an ADGF that prefers AMP. It is probable that other ADGF members prefer AMP as substrate as well. Therefore annotation of a protein by sequence identity as a member of the ADGF family does not necessarily indicate that its preferred substrate is adenosine.

We suggest that the inability of the human and Aplysia ADGFs to deaminate AMP is not due to the positioning of the catalytic residues but rather may reflect a steric clash and/or charge repulsion of the AMP phosphate. The HPAMPD has the same positioning of the catalytic residues as human ADGF. Clearly the same chemistry is taking place for deamination for all the ADGFs. The HPAMPD preference for AMP could be explained with a binding pocket that can accommodate the phosphate of AMP which contributes to substrate binding. Further work in this lab will attempt to elucidate the basis for the adenosine vs. AMP preference.

The ADGFs are involved in purinergic signaling. They are characterized as secreted growth factors and affect growth by modulating the extracellular levels of adenosine. We have shown that HPAMPD has an encoded signal peptide. Therefore, it is likely secreted in *H. pomatia*, and its site of action is extracellular. We propose that HPAMPD affects the level of extracellular AMP and is involved in purinergic signaling in some as yet unknown role. Skaldin [28] reported a bacterial ADGF homologue which is secreted and deaminates adenosine and suggested that the ADGFs evolved from a secreted bacterial homologue with a similar role involved in bacterial communication.

The occurrence of purinoreceptors in primitive invertebrates suggests that ATP is an early evolutionary extracellular signaling molecule [29, 30]. Perhaps *H. pomatia* uses an alternative pathway for modulating extracellular levels of ATP. For example, rather than converting ATP to AMP and then AMP to adenosine and deaminating adenosine to inosine as a last step, *H. pomatia*, removes AMP directly by deamination to IMP by HPAMPD as a last step.

In conclusion, HPAMPD is an AMP deaminase that is assigned by sequence similarity to the adenosine deaminase-related growth factor family which suggests that there is a biological role of AMP deaminases as growth factors. Furthermore, HPAMPD is a prime example for illustrating that the assignment of an enzymatic activity based only on sequence similarity in absence of biochemical evidence is imprudent.

## Materials and methods

### Purification of native AMPD from *H. pomatia*

The *H. pomatia* were obtained from My Happy Snails (myhappysnails.com) and fed on romaine lettuce for 5 days. The foot muscle from 2 snails was dissected. 1 gram was stored at -70˚ C and was later used for RNA extraction and cDNA cloning. The remaining 5 grams was homogenized in 20 ml of 20 mM Tris-HCl pH 7.5, 1 mM EDTA, 1 mM DTT, 1M NaCl. The homogenate was clarified by centrifugation and NaCl concentration was reduced to 200 mM by dilution with 20 mM Tris-HCl pH 7.5, 1 mM EDTA, 1 mM DTT. The clarified extract contained 600 AMP deaminase units.

The clarified extract was applied to a 5 ml Q sepharose column where 80% of the AMP deaminase flowed through the column. The Q flow-through was applied to a 5 ml HiTrap$^{TM}$ Heparin column. A gradient of NaCl was run to 1.2 M and the AMPD activity eluted at about 0.4M NaCl. The peak of activity was pooled and diluted to 0.1 M NaCl and subsequently applied to a 5 ml HiTrap$^{TM}$ Q column in 20 mM Tris-HCl pH 7.5, 1 mM EDTA, 1 mM DTT, 0.1 M NaCl. The activity flowed through the column and was then applied directly onto a 5 ml HiTrap$^{TM}$ SP column. A gradient to 1 M NaCl was run and the activity eluted at about 0.5 M NaCl. An aliquot (10 µL) of a peak fraction was subjected to *de novo* peptide sequence analysis. The remainder of the AMP deaminase active fractions were pooled and represent the native enzyme preparation with a yield of 30% at 200 units with a specific activity of about 160 units/ mg protein.

### AMPD assay

The assay is based on Kalckar's spectrophotometric method [10] where the decrease in optical density at 265 nM is measured. The reaction buffer is 20 mM Tris-HCl @ pH 7.5, 50 mM NaCl and typically the substrate concentration is 100 µM AMP. The reaction volume ranged from 200 to 400 µL. The optical density was determined with a Spectramax M3. One unit of the enzyme activity is defined as the amount of the enzyme which deaminates 1 µmole of 100 µmolar 5' AMP/min in 20 mM Tris-HCl pH 7.5, 50 mM NaCl at 25˚ C.

### *H. pomatia* transcriptome

The *H. pomatia* snails were obtained from a farmers' market in Germany and stored at -80˚ C. Approximately 100 mg of *H. pomatia* body were frozen in liquid nitrogen and ground to a powder. Total RNA was purified from rehydrated powder with the RNeasy Plant Mini Kit (Qiagen, Chatworth, CA). An mRNA transcript library for Illumina sequencing was created using the NEBNext rRNA Depletion Kit (Human/Mouse/Rat) (New England Biolabs, Ipswich, MA). A NextSeq 500 sequencer (Illumina, San Diego, CA) was used to do paired-end deep sequencing (2X 150 bp) of the library. The utility CutAdapt [31] was used to remove adapter sequences from the raw reads. The trimmed reads were then assembled using the Trinity software package [32]. The assembled transcriptome data was deposited at DDBJ/EMBL/GenBank under the accession GKIM00000000. The version described in this paper is the first version, GKIM01000000. Raw reads were deposited in the NCBI Sequence Read Archive (accession #PRJNA936131).

### Peptide sequencing by 2D LC-MS/MS

The purified HPAMPD protein fraction was digested in solution in parallel, with either trypsin at a ratio of trypsin to protein at 1:50 or subtilisin at a ratio of 1:160. The subtilisin digested fraction was analyzed twice, to include charge state 1 peptides. The trypsin digest was analyzed

with one injection. Each digest was directly injected onto an analytical C18 column and analyzed with Easy n1000 nLC-LTQ Orbitrap (Thermo Fisher Scientific, Waltham, MA) at 300 nL/min. All MS/MS spectra were analyzed with ProteomeDiscoverer version: 2.0.0.802 in one workflow. The fasta database consisted of open reading frames translated from the *H. pomatia* transcriptome. Peptide scoring validation was based on q<0.01 [12].

### RNA isolation for cDNA

The source of *H. pomatia* for cDNA was the same as for the HPAMPD enzyme preparation. About 1 gram of frozen *H. pomatia* foot tissue was pulverized in liquid nitrogen with mortar and pestle. About 0.2 grams of powdered tissue was homogenized in 1 ml of TRIzol reagent (Thermo Fisher). The debris was pelleted, and 0.2 ml of chloroform was added per ml of supernatant vortexed for 30 sec and phases separated by a 15 min centrifugation. 1 volume of ethanol was added to the aqueous layer and the mixture was applied to and eluted from a Directzol column (Zymo RNA miniprep kit) yielding 25 µg of RNA.

### cDNA cloning

First strand cDNA was generated from *H. pomatia* RNA with Protoscript II (NEB) and random primers. The first strand cDNA was used as a template for PCR amplification using primers A and B (see below) which were designed based on the transcriptome derived sequence for HPAMPD. The resulting 1556 bp fragment was inserted into the pET28c vector digested with EcoRI and NcoI using the NEBuilder HiFi Assembly kit (NEB).

Primer sequences:

A: `ACT TTA AGA AGG AGA TAT ACC ATG CTC GCC GTC AGA TTT CCG`
B: `AAG CTT GTC GAC GGA GCT CGT ACC GAT AAC TTT CTT GTT GGT TAG`

### Construction of expression cassette for *P. pastoris*

pD912-P$_{GAP}$ plasmid was obtained from ATUM Bio (Newark, CA). A linear expression cassette of pD912-P$_{GAP}$-Hp AMPD-His$_{10}$ was assembled using the NEBuilder HiFi DNA Assembly Cloning Kit (NEB #E5520S). Primers were designed using the NEBuilder Assembly Tool. The HPAMPD fragment was amplified with the forward primer C and reverse primer D (containing a 10X His-tag sequence) using plasmid pET28c-Hp AMPD as template. The pD912-P$_{GAP}$ fragment containing the transcriptional terminator (TT), ILV5 promoter, and zeocin resistant gene was amplified with the forward primer E and reverse primer F using pD912-P$_{GAP}$ plasmid as template. All products were purified by gel extraction. The linear expression cassette was then amplified with forward primer G and reverse primer H. The gel extracted product was used to transform electrocompetent *P. pastoris* D aox1 (MutS) cells (ATUM Bio.) and plated on yeast peptone dextrose agar medium supplemented with 1 M sorbitol and 500 µg/mL Zeocin (Teknova, Hollister, CA.).

Eight to twelve transformants were patched onto fresh selection plates and incubated for an additional 1–2 days at 30˚ C. For the identification of transformants by PCR, genomic DNA was isolated from each strain using lithium acetate/sodium dodecyl sulfate (LiOAc/SDS) method [33]. PCR was used to identify transformants having an integrated expression cassette.

Primer sequences:

C: `CGA GAA AAG AGA GGC CGA AGC TCT CGC CGT CAG ATT TCC G`
D: `TTG AGC GGC CGC CCC TTC AAC CTC AGT GGT GGT GGT GGT GGT GGT GGT GGT GGT GAC CGA TAA TTT CTT GTT GGT TAG`
E: `CAC CAC CAC CAC CAC CAC CAC CAC CAC CAC TGA GGT TGA AGG GGC GGC CGC`

```
F: AGC TTC GGC CTC TCT TTT C
G: GCT CAT TCC AAT TCC TTC TAT TAG
H: GAG CTC CAA TCA AGC CCA ATA AC
```

## *P. pastoris* clone harboring HPAMPD cDNA

The *P. pastoris* transformants that had integrated the expression cassette coding for the HPAMPD were grown in liquid cultures at 30˚ C in 5 mL of BMGY-Buffered Glycerol Complex Medium (Teknova) (1% yeast extract, 2% tryptone, 1.34% yeast nitrogen base without amino acids with ammonium sulfate, 0.0004% biotin, 1% glycerol as the carbon source, 100 mM potassium phosphate, pH 6.0). After 48 hours, the spent culture media was harvested. The culture fluid from 4 cultures one of which was a non-transformant control culture were each concentrated about 10-fold and assayed for AMP deaminase activity. One transformant had no detectable deaminase activity while the other two converted about 25% and 90% of the AMP to IMP in 4 minutes at 25˚ C.

## Purification of recombinant HPAMPD

The culture identified with the highest HPAMPD activity was then grown in 2400 ml of Tolner media [34] with 1% glycerol and 100 μg/ml ampicillin at 30˚ C for 72 hours. The culture was clarified by centrifugation and the supernatant culture fluid was brought 75% ammonium sulfate saturation and centrifuged to clarify. The purification was performed at 4˚ - 10˚ C except where noted. The 3.5-liter supernatant was then diluted with 700 ml of water and applied to a 250 ml bed Phenyl Sepharose column. After loading the column with supernatant, the column was washed with 500 ml of 3M ammonium sulfate, 20 mM Tris-HCl pH 7.5, 1mM EDTA and 1 mM DTT. A reverse gradient of 1800 ml from 2M to 0 M ammonium sulfate was applied to the column. The HPAMPD activity that eluted from the column was pooled (120 ml) and diluted with 2 liters of 20 mM Tris-HCl pH 7.5, 25 mM NaCl, 1mM EDTA, 1 mM DTT. This was applied to a 10 ml Heparin HyperD column washed with 20 mM Tris-HCl pH 7.5, 1 mM EDTA, 1 mM DTT, 50 mM NaCl and eluted with a 400 ml gradient from 50 mM to 1 M NaCl in 20 mM Tris-HCl pH 7.5, 1 mM EDTA, 1 mM DTT. The peak of activity was pooled and half of the pool was treated with Endo Hf (NEB) by first dialyzing the 25 ml against 2 changes of 500 ml of 20 mM Na acetate pH 6.0, 1 mM EDTA, 50 mM NaCl, 1 mM DTT. Five hundred thousand units of EndoHf was added to the 25 ml dialysate and incubated at 25˚ C overnight. A small amount of precipitate formed and was pelleted and removed by centrifugation. In order to remove the EndoHf from the HPAMPD the supernatant was brought to 100 mM Tris -HCl pH 7.5 and 100 mM NaCl and applied to a one ml amylose resin (NEB) column at a rate of about 1 drop per second at 25˚ C. The flow through containing the HPAMPD was then applied directly to a 5 ml HiTrap$^{TM}$ Q column in line with a 5 ml HiTrap$^{TM}$ SP column and washed with 20 mM Tris-HCl pH 7.5, 1mM EDTA and 1 mM DTT, 50 mM NaCl. The Q column was removed and a 50 ml gradient from 50 mM to 1M NaCl was applied to the SP column. The peak of HPAMPD activity was pooled and dialyzed against 50% glycerol, 10 mM Tris-HCl pH 7.4, 1 mM DTT, 0.1 mM EDTA. The recombinant HPAMPD preparation is stored at -20˚ C. The protein is >95% homogenous by PAGE (S1 Fig in S1 File) and protein concentration is 2.6 mg/ml based on $OD_{280}$ with an extinction coefficient of 85,000 $M^{-1}$ $cm^{-1}$.

## Human and *A. californica* ADGF

The human ADGF (Recombinant Human Adenosine deaminase 2/CECR1) #7518-AD-010 was from R&D Systems, Minneapolis, MN and the *A. californica* ADGF was purified from construct described below. The *P. pastoris* clone carrying the *A. californica* ADGF cDNA was

grown and purified as described for the *H. pomatia* construct except deglycosylation was not performed.

## Yeast culture conditions and expression

Construction of the *A. californica* construct for expression in *P. pastoris* was performed as follows: The transformation cassette for *A. californica* ADGF, was generated by PCR using primers 1941 and 1942 and as a template the clone in PD912-GAP which was obtained from Genscript, Piscataway, NJ. The PCR product was confirmed for size by agarose gel electrophoresis and subsequently purified by QIAquick (Qiagen) columns. The purified product (1 μg) was electroporated into *P. pastoris* competent cells and positive colonies identified as described above.

Primer sequences:
1941: `GCT CAT TCC AAT TCC TTC TAT TAG`
1942: `GAG CTC CAA TCA AGC CCA ATA AC`

## Supporting information

**S1 File.**
(DOCX)

**S1 Table. List of identified peptides from MS analyses of the digests.**
(XLSX)

**S2 Table. List of correspondence of transcriptome contig numbers to protein IDs.**
(XLSX)

## Acknowledgments

We thank Sean Johnson for a critical reading of the manuscript, the DNA sequencing core lab (NEB), Saulius Vainauskas for advice with *Pichia*, Vladimir Potapov for help with data deposition, Tom Evans, Richard Roberts and Donald Comb for encouraging basic research at New England Biolabs.

## Author Contributions

**Conceptualization:** Ira Schildkraut.

**Data curation:** Mehul B. Ganatra, Cristian Ruse, Christopher H. Taron.

**Formal analysis:** George Tzertzinis, Mehul B. Ganatra, Cristian Ruse, Ira Schildkraut.

**Investigation:** George Tzertzinis, Mehul B. Ganatra, Cristian Ruse, Bryce Causey, Liang Wang, Ira Schildkraut.

**Supervision:** Ira Schildkraut.

**Writing – original draft:** Ira Schildkraut.

**Writing – review & editing:** George Tzertzinis, Mehul B. Ganatra.

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
