## [Decision Letter · Decision Letter 0]

29 Jun 2023

PONE-D-23-13161The AMP deaminase of the mollusk Helix pomatia is an unexpected member of the adenosine deaminase-related growth factor (ADGF) familyPLOS ONE

Dear Dr. Schildkraut,

Thank you for submitting your manuscript to PLOS ONE. After careful consideration, we feel that it has merit but does not fully meet PLOS ONE’s publication criteria as it currently stands. Therefore, we invite you to submit a revised version of the manuscript that addresses the points raised during the review process.

We look forward to receiving your revised manuscript.

Kind regards,

Israel Silman

Academic Editor

PLOS ONE

Journal Requirements:

Additional Editor Comments :

Although the two reviewers view the manuscript favourably, they both have some comment that you are requested to address as fully as possible in your revised manuscript. 

Reviewers' comments:

Reviewer's Responses to Questions

**Comments to the Author**

1. Is the manuscript technically sound, and do the data support the conclusions?

Reviewer #1: Yes

Reviewer #2: Yes

2. Has the statistical analysis been performed appropriately and rigorously? 

Reviewer #1: Yes

Reviewer #2: Yes

3. Have the authors made all data underlying the findings in their manuscript fully available?

Reviewer #1: Yes

Reviewer #2: Yes

4. Is the manuscript presented in an intelligible fashion and written in standard English?

Reviewer #1: Yes

Reviewer #2: Yes

5. Review Comments to the Author

Reviewer #1: Tzertzinis et al. have discovered a new member of the adenosine deaminase growth factor (ADGF) family that deaminates AMP rather than adenosine. This finding showcases the remarkable adaptability of the ADGF fold for different extracellular substrates. Moreover, it emphasises the broader significance of these secreted enzymes in cell-cell signalling. The experimental procedures conducted in this study are robust, yielding reliable results. The clarity and coherence of the paper's presentation are commendable. However, I would like to offer a few comments and suggestions:

1. Page 2, line 48. "This subfamily of adenyl deaminase is represented in humans as the enzyme encoded by ADA2 (also known as CECR1) gene" Please, cite the paper, in which CECR1 was identified as ADA2:

Andrey V Zavialov & Ake Engström "Human ADA2 belongs to a new family of growth factors with adenosine deaminase activity" Biochem J. 2005 Oct 1;391(Pt 1):51-7. doi: 10.1042/BJ20050683.

2. Page 6, lines 129-134. Please change "identity score" to "protein sequence identity"

3. Page 6, line 133. Please cite here the paper that described the ADA2 structure:

Anton Zavialov et al. J Biol Chem. 2010 Apr 16;285(16):12367-77. doi: 10.1074/jbc.M109.083527.

4. Page 7. In human ADA2, there are several structural elements that are responsible for its dimerisation (Zavialov et al. JBC 2010). Are they predicted by AlphaFold2 in HPAMPD? Is HPAMPD a dimer?

5. Page 10. Chapter: Comparison of A. California and human ADGF ...

It has been known for many years that human ADA2 does not deaminate AMP. Although the gene of ADA2 was not known at that time, its activity was throughly examined:

J G Niedzwicki & D R Abernethy, Structure-activity relationship of ligands of human plasma adenosine deaminase 2. Biochem Pharmacol. 1991 Jun 1;41(11):1615-24. doi: 10.1016/0006-2952(91)90162-x.

Please, change the text accordingly and cite the paper.

Reviewer #2: The authors present an intriguing manuscript showing AMP deaminase is a member of the adenosine deaminase-related growth factor in Helix pomatia, which deaminates adenosine 5' monophosphate in preference to adenosine.

The following are my comments and critique that should be addressed before acceptance for publication:

1. The figure 6 in your paper is a bit blurry. Please consider replacing it with a clearer one.

2. Add space between words and values/numbers, check your paper carefully, please.

3. Line 201. Replace “54 ° C” with “54 °C”.

4. Line 178. Please complete this sentence and cite the paper you use properly.

5. Line 225. Could you please provide any data or explanation to support this conclusion?

6. Line 238-240. How to explain the conclusion: some fraction of the 1777 ADGF members are incorrectly annotated?

6. PLOS authors have the option to publish the peer review history of their article (what does this mean?). If published, this will include your full peer review and any attached files.

Reviewer #1: **Yes: **Anton Zavialov

Reviewer #2: No

---

## [Author Response · Author response to Decision Letter 0]

6 Jul 2023

Respone to Reviewers file was uploaded

---

## [Editor Report · Decision Letter 1]

10 Jul 2023

The AMP deaminase of the mollusk Helix pomatia is an unexpected member of the adenosine deaminase-related growth factor (ADGF) family

PONE-D-23-13161R1

Dear Dr. Schildkraut,

We’re pleased to inform you that your manuscript has been judged scientifically suitable for publication and will be formally accepted for publication once it meets all outstanding technical requirements.

Kind regards,

Israel Silman

Academic Editor

PLOS ONE
---

## [Editor Report · Acceptance letter]

12 Jul 2023

PONE-D-23-13161R1 

The AMP deaminase of the mollusk *Helix pomatia* is an unexpected member of the adenosine deaminase-related growth factor (ADGF) family 

Dear Dr. Schildkraut:

I'm pleased to inform you that your manuscript has been deemed suitable for publication in PLOS ONE. Congratulations! Your manuscript is now with our production department. 

Kind regards, 

on behalf of

Prof. Israel Silman 

Academic Editor

PLOS ONE